# Dynamic Multimodal Evaluation via Knowledge-Enhanced Benchmark Evolution

**Junzhe Zhang** [1]  **Huixuan Zhang** [1]  **Xiaojun Wan** [1]

## Abstract

The rapid development of multimodal large language models (MLLMs) has created an urgent demand for more reliable and robust evaluation protocols, however, existing static benchmarks are prone to data contamination and performance saturation, which can result in inflated or misleading evaluation results. To address these limitations, we first introduce a graph formulation to represent both static and dynamic visual question answering (VQA) samples. Building upon this formulation, we propose Knowledge-Enhanced Benchmark Evolution (KBE), a dynamic multimodal evaluation framework that first analyzes the original static benchmark, then expands it by integrating multimodal knowledge, transforming the static benchmark into a controllable, dynamic evolving version. Crucially, KBE can both reconstruct questions by Re-selecting visual information in the original image and expand existing questions with external textual knowledge. By explicitly controlling the degree of question exploration, KBE enables difficulty-controllable evaluation across a wide range of model capabilities. Extensive experimental results demonstrate that KBE effectively mitigates data contamination and benchmark saturation, while providing a more comprehensive and flexible assessment of MLLM performance.

## 1. Introduction

Multimodal large language models(MLLMs) have been developing rapidly in recent years, demonstrating remarkable performance across a wide range of tasks(Li et al., 2024b; Bai et al., 2025). This rapid progress has motivated the cre-

[1] Peking University, Wangxuan Institute of Computer Technology, Beijing, China. Correspondence to: Xiaojun Wan <wanxiaojun@pku.edu.cn>.

*Proceedings of the $43^{rd}$ International Conference on Machine Learning*, Seoul, South Korea. PMLR 306, 2026. Copyright 2026 by the author(s).

ation of an increasing number of multimodal benchmarks designed to evaluate their capabilities. Several traditional static benchmarks(Marino et al., 2019; Schwenk et al., 2022) have been carefully curated with comprehensive testing coverage and rigorous construction processes. These benchmarks provide evidence of how current multimodal models perform across diverse multimodal tasks, and are crucial in understanding the strengths and weaknesses of MLLMs.

However, current evaluation practices also suffer from several implicit limitations. A primary concern lies in the risk of data contamination(Song et al., 2025). Most of the aforementioned open-source benchmarks release their test samples and labels to facilitate reproducibility of comparison across different multimodal large models. While this openness ensures transparency, it also increases the risk that open-sourced test data may be inadvertently leaked into training corpora of existing MLLMs. The wide availability of benchmark test sets means that portions of these datasets can be unintentionally included during large-scale pretraining. As a result, the reliability of static open-source benchmarks gradually diminishes over time, since their effectiveness as unbiased evaluators is compromised by potential overlap with training data. Another issue lies in data saturation. As multimodal large language models continue to develop rapidly, their performance on many established benchmarks keeps improving. However, the difficulty of static benchmarks remains fixed and cannot evolve along with the increasing capabilities of newer MLLMs. As a result, certain models have already achieved high scores on some widely used datasets, raising concerns about the diminishing discriminative power of such benchmarks. In this scenario, benchmarks that were once sufficiently challenging can no longer provide a reliable separation between state-of-the-art systems, thereby limiting their utility in driving future progress.

To address these concerns, a conventional approach is to constantly design new benchmarks whose difficulty is suitable for evaluating the current capabilities of MLLMs. Such new constructed benchmarks can temporarily control task difficulty and mitigate the risks of data contamination. However, the validity of these benchmarks inevitably diminishes over time, and the repeated construction of specialized static datasets requires substantial human effort and is time-

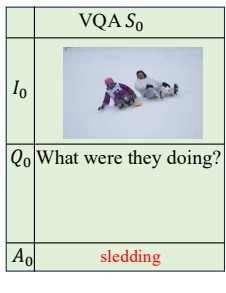
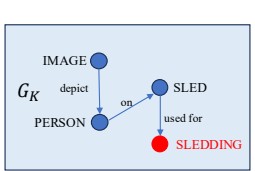
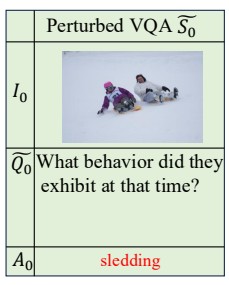
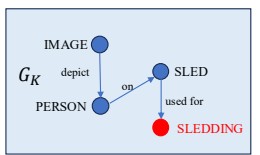
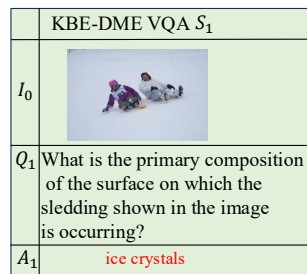
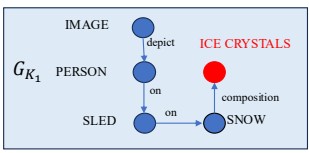

*Figure 1.* Figure of the original VQA test sample $S_0$, the perturbed test sample $\tilde{S}_0$ generated by perturbation methods, and the dynamically generated test sample $S_1$ produced by KBE-DME. Here, $G_K$ denotes the key information subgraph extracted from a VQA test sample. As shown, the perturbed VQA test sample does not alter the $G_K$ of the question.

consuming. We argue that a more sustainable solution is to develop frameworks that enables benchmarks to evolve with MLLMs, enabling dynamic evaluation. Only through such adaptive benchmarks can we overcome the dual limitations of data contamination and data saturation inherent in the traditional evaluation with static benchmark.

There are some multimodal dynamic evaluation methods(Yang et al., 2025) based on perturbation. However, the perturbed VQA test sample sometimes does not alter the core of the question as shown in Figure 1. To overcome the challenges of data contamination and data saturation, we introduce KBE-DME: Dynamic Multimodal Evaluation via Knowledge-Enhanced Benchmark Evolution. Our approach starts with the multimodal tasks represented in the standard VQA format, and further modeling each VQA using Graph formulation with mutlimodal knowledge triplets. For each question, we extract a set of candidate multimodal knowledge triplets and then identify the subset composed of key triplets that are necessary for answering the question. Based on this representation, KBE-DME introduces a framework dynamically evolves benchmarks in two different ways: 1.Re-selection of key triplets from the pool of candidate multimodal knowledge triplets; 2.Exploration with external knowledge triplets, where new triplets are incorporated into the key triplets set to enrich the required knowledge. By integrating the modified key triplets with the information of original VQA sample, KBE-DME synthesizes novel VQA questions with controllable difficulty. These dynamic questions can continuously evolve with the progress of multimodal large models through our framework. This dynamic construction not only mitigates the risks of test set leakage but also ensures that the difficulty of evaluation adapts to the improving capabilities of MLLMs.

Our proposed method, KBE-DME, exhibits strong generalization ability and can be applied to transform different multimodal tasks of VQA format for dynamic evaluation. We apply KBE-DME on two widely used benchmarks, OK-VQA(Marino et al., 2019) and A-OKVQA(Schwenk et al., 2022), and evaluate five representative MLLMs. Through extensive experiments and analyses of the generated dynamic test data, our results demonstrate that KBE-DME is capable of dynamically constructing test sets of various difficulty levels based on static benchmarks. This enables a dynamic and difficulty-controllable evaluation of MLLMs, effectively overcoming the limitations of traditional static benchmarks.

Our contributions are[1]:

- We introduce a novel graph formulation to represent a VQA sample, where the multimodal knowledge is represented as triplets. Within this formulation, the process of dynamic test data generation for evaluation can be naturally expressed as transformations of triplets in the graph structure.

- We propose KBE-DME, a dynamic evaluation framework which evolves static multimodal benchmarks via re-selection and exploration strategy, generating difficulty-controllable test data that co-evolves with the progress of MLLMs.

- We conduct extensive experiments with KBE-DME on two static VQA benchmarks, evaluate five representative MLLMs and perform detailed analyses of the dynamically generated data. The results demonstrate

---

[1]Our code is available at https://github.com/reroze/KBE-DME

*Table 1.* Comparisons of current dynamic evaluation methods, NPHardEval(Fan et al., 2024), DyVal(Zhu et al., 2024a), MPA(Zhu et al., 2024b) and VLB(Yang et al., 2025).* means that although the DyVal paper includes a preliminary study on sentiment classification, their framework still cannot be directly applied to broad non-reasoning generation tasks. Dyn-VQA(Li et al., 2025) is different from all these methods, we will discuss it separately.

| Methods | Multimodal | Task Generalize | Not Adversarial Methods | External Knowledge | Difficulty Control |
|---|---|---|---|---|---|
| **NPHardEval** | ✗ | ✗ | ✓ | ✗ | Algorithm-depended |
| **DyVal** | ✗ | ✗* | ✓ | ✗ | Fine-grained |
| **MPA** | ✗ | ✓ | ✗ | ✓ | Coarse-grained |
| **VLB** | ✓ | ✓ | ✗ | ✗ | Coarse-grained |
| **KBE-DME** | ✓ | ✓ | ✓ | ✓ | Fine-grained |

that KBE-DME enables high-quality and difficulty-controllable dynamic evaluation of MLLMs.

## 2. Related Work

### 2.1. Static Multimodal Evaluation

Static multimodal evaluation has long served as the standard paradigm for assessing the capabilities of multimodal models. Early efforts introduced benchmarks such as MSCOCO Captions(Chen et al., 2015), VQA-v2(Goyal et al., 2017), OK-VQA(Marino et al., 2019), and A-OKVQA(Schwenk et al., 2022), which provided fixed datasets and standardized metrics to measure abilities such as visual understanding, reasoning, and image–text alignment. These static benchmarks played a crucial role in model developing progress by offering a common ground for model comparison. With the rapid advancement of multimodal large models (MLLMs) in recent years(Li et al., 2024b; Bai et al., 2025), previous proposed benchmarks are no longer sufficient to meet the need for evaluating increasingly powerful MLLMs. To keep pace with increasingly powerful models, plenty of new static benchmarks have been proposed, aiming to provide broader coverage and more challenging tasks. MMBench(Liu et al., 2024), MME(Fu et al., 2026), MMStar (Chen et al., 2024) and SEED-Bench(Li et al., 2024a) provide comprehensive multi-dimensional evaluations like reasoning, OCR, and others. Despite their success, static multimodal benchmarks remain inherently constrained by fixed difficulty and potential data contamination once released. Although some studies have attempted to change their evaluation questions(Shah et al., 2019; Gokhale et al., 2020), these methods are typically designed for specific datasets and are difficult to serve as a widely applicable dynamic evaluation strategy for other multimodal static benchmarks.

### 2.2. Dynamic Evaluation

To mitigate the data contamination and data saturation issues, recent studies have explored dynamic evaluation(Jiang et al., 2025; Yang et al., 2025), where test data are perturbed(Yang et al., 2025) or regenerated(Jiang et al., 2025)

to adapt difficulty and reduce data contamination effects. In the field of text-only dynamic evaluation, DyVal(Zhu et al., 2024a) dynamically generate test samples to mitigate data comtamination. NPHardEval(Fan et al., 2024) generate new samples for NP-hard math problems evaluation. MPA(Zhu et al., 2024b) apply agent to generate new evluation samples.

However, in the multimodal domain, research on dynamic evaluation remains relatively limited. VLB(Yang et al., 2025) represents one of the first attempts to bootstrap both images and text simultaneously by editing objects or backgrounds in images, replacing or rephrasing words in questions, and adding related or unrelated textual content to perturb the original VQA problems. Liu & Zhang (2025) proposes a multimodal dynamic evaluation framework to perturb the multimodal task itself instead of perturbing inputs. While perturbation-based methods indeed modify the test inputs, their impact is relatively limited compared to regenerating entirely new test data. Moreover, perturbation approaches offer little control over the difficulty level of the generated data. To achieve both a broader range of dynamically generated data and finer-grained difficulty control, we propose our dynamic multimodal evaluation framework, KBE-DME. The difference between our method and previous works can be seen in Table 1.

## 3. KBE-DME

### 3.1. Graph Formulation

We represent a multimodal VQA problem using a graph formulation, where each problem is abstracted as a structured graph composed of multiple multimodal knowledge triplets.

**Knowledge Triplet**  A unit of knowledge can be represented as a multimodal knowledge triplet $(s, r, o)$. $(s, r, o)$, where $s$ denotes the subject of the triplet, $r$ specifies the relation, and $o$ corresponds to the object associated with $s$ under relation $r$.

**Graph Definition of Static VQA**  We treat the $s$ and $o$ in a knowledge triplet as nodes in the graph, and use the

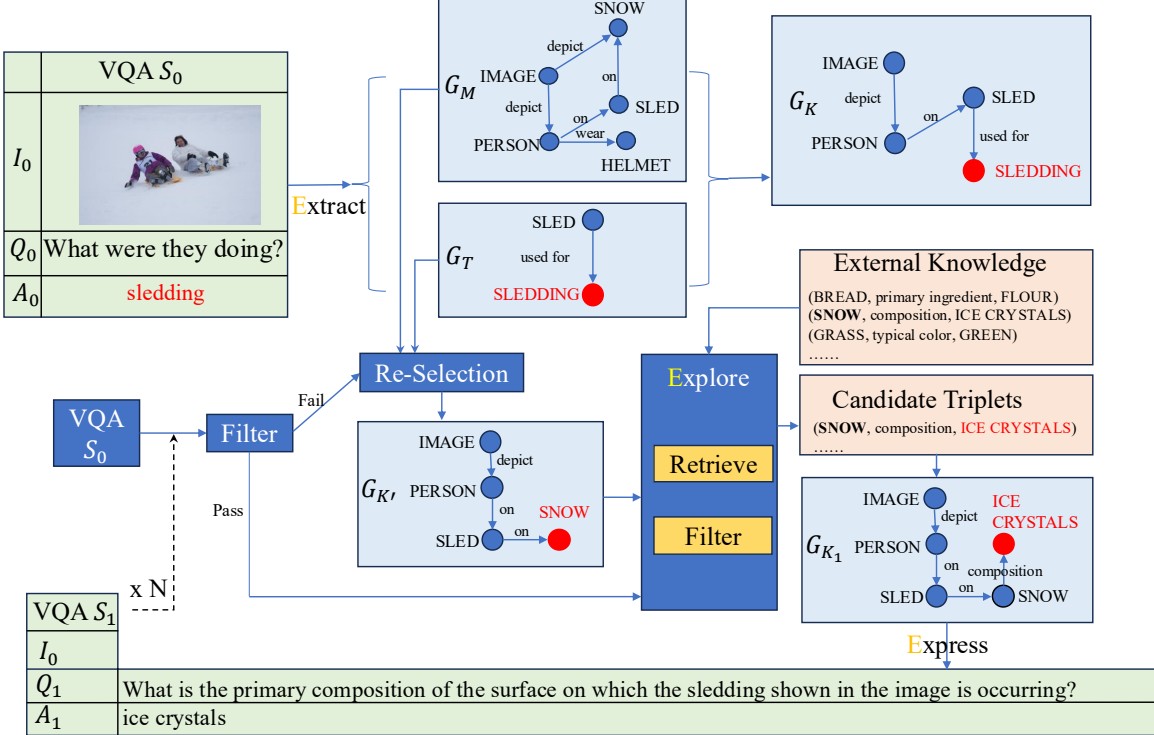

*Figure 2.* Figure of our Graph Formulation and the KBE-DME framework. The upper part of the figure uses a static VQA sample $S_0$ to exemplify our graph representation of a VQA problem, while the lower part demonstrates the KBE-DME framework for dynamically constructing VQA test data.

corresponding triplet $(s, r, o)$ as an directed edge connecting $s$ and $o$. We first construct visual knowledge triplets $M = \{(s_m, r_m, o_m)\}$ to represent the visual information of a VQA sample, and then construct textual knowledge triplets $T = \{(s_t, r_t, o_t)\}$ to represent the worldwide textual background knowledge of the VQA sample. Based on this representation, we construct a Visual Graph $G_M = <V_M, E_M>$ with visual knowledge triplets, which is:

$$V_M = \{s_m, o_m \mid \exists (s_m, r_m, o_m) \in M\} \quad (1)$$

$$E_M = \{(s_m, o_m, [r_m]) \mid \exists (s_m, r_m, o_m) \in M\} \quad (2)$$

and a Textual Graph $G_T = <V_T, E_T>$ with textual knowledge triplets, which is:

$$V_T = \{s_t, o_t \mid \exists (s_t, r_t, o_t) \in T\} \quad (3)$$

$$E_T = \{(s_t, o_t, [r_t]) \mid \exists (s_t, r_t, o_t) \in T\} \quad (4)$$

Note that we formulate the edge set $E$ as a edge $(s, o, [r])$ with property $r$ to handle the relation between $s, o$.

The nodes in the visual subgraph $G_M$ are composed of the subjects and objects from the visual knowledge triplets $M$, and each visual knowledge triplet corresponds to a directed edge in the graph. The textual subgraph $G_T$ is constructed in the same manner. A concrete example is illustrated in Figure 2.

However, answering a VQA question typically does not require utilizing all the multimodal information ($G_M$ and $G_T$) contained in the sample. Instead, only a subset of key information($G_K$) is necessary to achieve the correct answer. Based on this observation, we extract the key knowledge triplets $K = \{(s_k, r_k, o_k)\} \subseteq M \cup T$ that are essential for answering the VQA question from the set of visual knowledge triplets $M$ and textual knowledge triplets $T$. These triplets are then used to construct a new key multimodal knowledge subgraph $G_K = <V_K, E_K>$, which is:

$$V_K = \{s_k, o_k \mid \exists (s_k, r_k, o_k) \in K\} \quad (5)$$

$$E_K = \{(s_k, o_k, [r_k]) \mid \exists (s_k, r_k, o_k) \in K\} \quad (6)$$

The role of $G_K$ is similar to a graphical representation of the rationale required to answer a VQA question.

At this stage, we obtain a graph representation for each static VQA problem $S_0 = \{I_0, Q_0, A_0\}$, $S_0$ denotes the original data of a given test VQA sample, where $I_0$, $Q_0$, and $A_0$ represent the corresponding input image, input question, and answer, respectively. The potential multimodal knowledge contained in the problem is modeled as a visual subgraph $G_M$ and a textual subgraph $G_T$, while the key information required to answer the question is captured by the key subgraph $G_K$. Consequently, dynamically altering a VQA problem can be naturally formulated as dynamically modifying its corresponding key subgraph $G_K$.

$$S_0 = \{I_0, Q_0, A_0\} \sim \{G_M, G_T, G_K\} \quad (7)$$

**Graph Representation of Dynamic VQA** Intuitively, there are two ways to modify the key subgraph $G_K$ corresponding to a VQA sample.

**(1) Re-Selection**: by choosing a different set of key knowledge triples $K' = \{(s_{k'}, r_{k'}, o_{k'})\} \subseteq M \cup T \neq K$ from the existing visual triplets $M$ and textual triplets $T$, we can generate a new key subgraph $G_{K'} = <V_{K'}, E_{K'}>$. The formal expression is given as follows:

$$V_{K'} = \{s_{k'}, o_{k'} \mid \exists (s_{k'}, r_{k'}, o_{k'}) \in K'\} \quad (8)$$

$$E_{K'} = \{(s_{k'}, o_{k'}, [r_{k'}]) | \exists (s_{k'}, r_{k'}, o_{k'}) \in K'\} \quad (9)$$

The new VQA sample $S_0'$ can be then formulated as:.

$$S_0' = \{I_0, Q_0', A_0'\} \sim \{G_M, G_T, G_{K'}\} \quad (10)$$

**(2) External Knowledge Exploration**: by selecting appropriate knowledge triplets from external sources, we expand the original set of key triples $K$ into an extended set $K_n = \{(s_{k_n}, r_{k_n}, o_{k_n})\}$ with new textual triplets set $N$. Using $K_n$, we generate a new key subgraph $G_{K_n} = <V_{K_n}, E_{K_n}>$. Unlike Re-Selection, $T_n$ is expanded together with the augmentation of the triplets. The formal expression is as follows:

$$V_{K_n} = \{s_{k_n}, o_{k_n} \mid (s_{k_n}, r_{k_n}, o_{k_n}) \in K_n\} \quad (11)$$

$$E_{K_n} = \{(s_{k_n}, o_{k_n}, [r_{k_n}]) \mid \exists (s_{k_n}, r_{k_n}, o_{k_n}) \in K_n\} \quad (12)$$

$$T_n = N \cup T, K_n = N \cup K. \quad (13)$$

The new VQA sample $S_n$ can be formulated as shown above.

$$S_n = \{I_0, Q_n, A_n\} \sim \{G_M, G_{T_n}, G_{K_n}\} \quad (14)$$

As the corresponding graph structure is updated, the original VQA problem is transformed into a new one for evaluation. We measure the difficulty of the generated VQA problem based on the number of edges $|E_K|$ in its key subgraph $G_K$.

### 3.2. Knowledge Enhanced Benchmark Evolution

We represent each VQA sample following proposed graph formulation and model the dynamic evaluation process accordingly. In the following, we present our Dynamic Evaluation Framework. Our overall pipeline can be divided into three components: Extract, Exploration, and Express, which can be seen in Figure 2.

**Extract** We first perform information extraction based on the input image, question, and answer of the given VQA data, obtaining the corresponding visual knowledge triplets $M$ and textual knowledge triplets $T$. We then identify the key triplets $K$ that are required to answer the VQA question by combining the extracted triplets with the original VQA input. To achieve this, we employ the powerful and general-purpose MLLM as dynamic generation model. Once $M$, $T$, and $K$ are obtained, we construct the graph representations of the VQA sample, namely visual graph $G_M$, textual graph $G_T$, and key subgraph $G_K$ according to Eq (1-6). We believe that for a standard VQA sample, its corresponding $G_K$ should contain at least one edge from $G_M$, i.e, at least one visual triplet. Otherwise, answering this VQA sample would not require any visual information, which we consider to be unreasonable. Therefore, we retain only results whose $G_K$ includes at least one edge from $G_M$.

**Explore** After obtaining $G_M$, $G_T$, and $G_K$ for an original question, we expand the original problem to generate new VQA questions. Specifically, we adopt two strategies for question expansion and generation: **Triplets Re-Selection** and **Triplets Exploration**.

To ensure the reliability of knowledge during the exploration process, we introduce a filtering step. We first perform an answer filtering step, which consists of three components: representativeness filtering, part-of-speech filtering, and cycle-check filtering. Representativeness filtering applies the MLLM to determine whether the corresponding triplet (s,r,o) is representative. Part-of-speech filtering examines the POS of the candidate answer. Specifically, we assume that answers with a noun POS are more suitable for further exploration. Finally, cycle-check filtering ensures that for newly expanded triplets, the output cannot be identical to any subject in the original key triplets, since this would introduce cycles in $G_K$, leading to unreasonble generated new question.

For a VQA sample $S_0$ from an existing dataset, we assume that representative filtering has already been considered

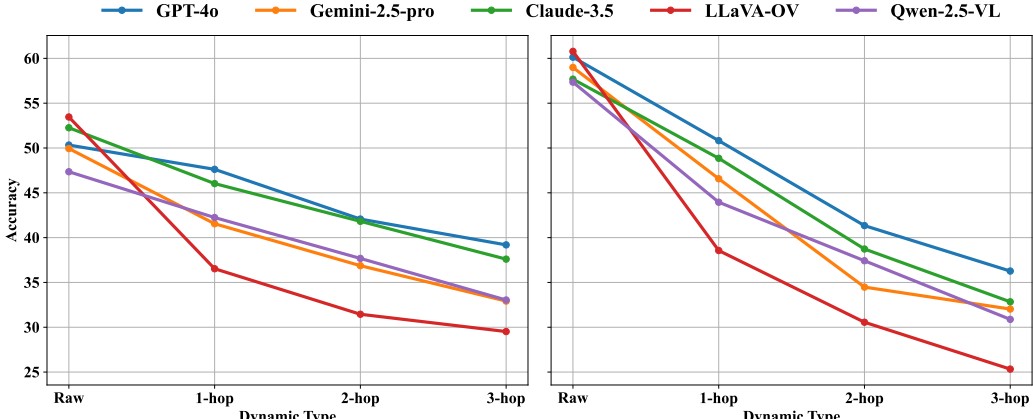

*Figure 3.* Main Results of five different MLLMs on original static benchmark(Raw) and our generated dynamic benchmark with exploration of different(1-3) hops through our KBE-DME framework. The left figure shows the results on OK-VQA, and the right figure shows the results on A-OKVQA. Specific result can be seen in Table 3 in Appendix B.

during its construction, and thus all original questions are regarded as passing this step. We then apply part-of-speech filtering to the original answer $A_0$, which divides the samples into two groups: $Pos_T 1$, where $A_0$ is a noun, and $Pos_F$, where it is not. For data belonging to $Pos_F$, we perform Re-Selection to construct new VQA questions whose answers are nouns, resulting in a new set $Pos_T 2$. Finally, $Pos_T 1$ and $Pos_T 2$ together form $Pos_T$, the collection of VQA questions whose answers all satisfy the noun constraint.

The implementation of Re-Selection is as follows. We first identify the image root node of $G_M$ which is named IMAGE. We then search within $G_M$ and $G_T$ for paths that include this image root node. Among these, we define as the valid path set those paths whose terminal node (the endpoint of the last edge) is a noun. Finally, we select the longest path from this valid set to serve as the new key graph $G_{K'}$. A concrete example is illustrated in Figure 2.

For the data in $Pos_T$, we perform knowledge exploration. Specifically, we first employ the dynamic generation MLLM to generate a set of candidate expandable knowledge triplets, where the subject s of each triplet corresponds to the current answer. We then apply our filtering strategies to these answer-related triplets. Finally, from the filtered triplets that meet the requirements, we randomly select one for knowledge exploration and incorporate it into the current problem's key subgraph to obtain new key subgraph $G_{K_1}$. Using the KBE-DME framework, we can iteratively repeat this process up to three times, thereby obtaining key subgraphs with different hop expansions, namely $G_{K_2}$ and $G_{K_3}$

**Express** We employ MLLM to transform the generated key subgraph into a new VQA question–answer pair. Taking the first expansion as an example, we provide MLLM with

the image, question, and answer of the current VQA sample $S_0$, along with its corresponding key subgraph $G_K$. We then specify the knowledge triplet to be expanded and designate the new VQA answer as the output of this triplet. Finally, MLLM is instructed to generate a new input question based on this information, thereby completing the transformation from the graph representation to a new VQA sample. An example can be seen in Figure 7 in Appendix A. Detailed prompts can be seen in Appendix J

## 4. Experiment

### 4.1. Experiment Setup

We choose OK-VQA(Marino et al., 2019) and A-OKVQA(Schwenk et al., 2022) as the primary static datasets for our experiments. We also conduct experiments on other datasets, more details can be found in the Appendix F. Specifically, we select the validation splits of these datasets. The validation sets of OK-VQA and A-OKVQA contain approximately 5k and 1.1k samples, respectively. After our filtering process, we obtain 2.6k samples from OK-VQA and 0.6k samples from A-OKVQA as the starting points for the static test sets in our dynamic evaluation. We apply GPT-4o as our dynamic generation model in our main experiment and conduct ablation study with applying Qwen2-VL-72B as dynamic generation model. More experimental details can be seen in Appendix E.

### 4.2. Main Results

We expand each original question up to three hops following the procedure illustrated in Figure 2. We then evaluate five multimodal large language models on the datasets obtained after expansion of different hops, with the results presented in the Figure 3.

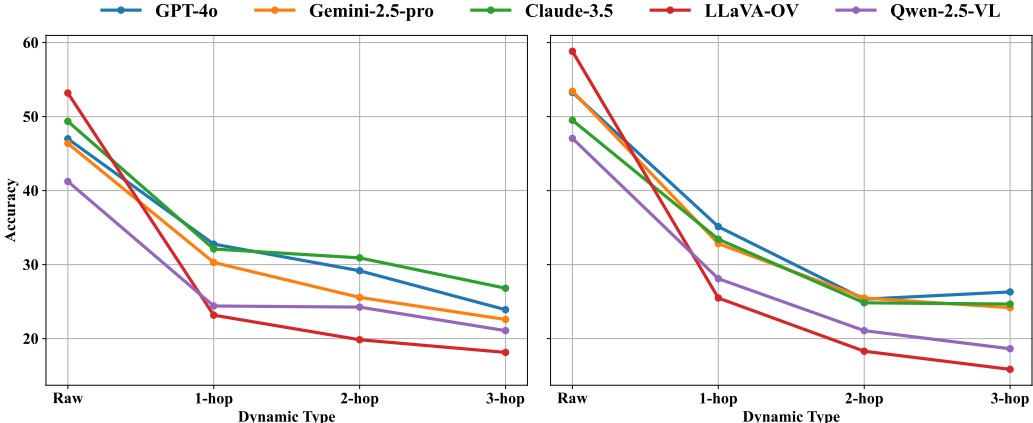

*Figure 4.* Main Results of five different MLLMs on original static benchmark(Raw) and our generated dynamic benchmark with exploration of different(1-3) hops through our KBE-DME framework with strict character matching as evaluation method. The left figure shows the results on OK-VQA, and the right figure shows the results on A-OKVQA. Specific result can be seen in Table 4 in Appendix B.

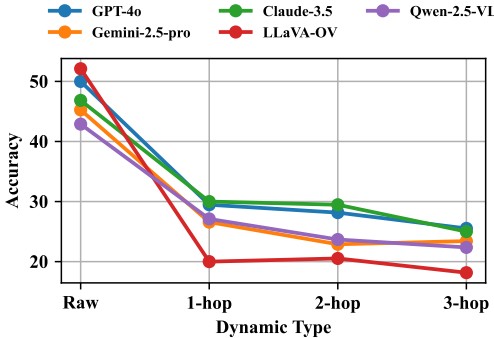

*Figure 5.* Main Results of five different MLLMs on original static benchmark(raw) and our generated dynamic benchmark with exploration of different(1-3) hops through our KBE-DME framework with Qwen2-VL-72B as Dynamic Generation Model. Specific result can be seen in Table 5 in Appendix B.

We observe that as the number of expansion hops increases, the performance of all five tested models declines across both datasets. This indirectly demonstrates that our dynamic evaluation framework provides reliable control over task difficulty.

In addition, we find that some models exhibit relatively smooth performance degradation across different expansion hops, such as GPT-4o, Claude-3.5, and Qwen-2.5-VL in the dynamic evaluation based on the OK-VQA dataset. However, for Gemini-2.5-pro and LLaVA-OV, the performance drop is more pronounced during the first expansion, while the decline becomes more gradual in subsequent hops. In the dynamic evaluation based on the A-OKVQA dataset, GPT-4o and Claude-3.5 again maintain relatively smooth degradation, whereas Qwen-2.5-VL shows a comparatively larger drop at the first expansion than at later ones.

The marginal effect makes it reasonable that the perfor-

mance gap between models diminishes as the test questions become more difficult. However, if a model exhibits a substantial performance drop at the very first expansion, this may indicate a potential risk of data contamination on the corresponding dataset. We conduct a preliminary analysis of the reasons behind the performance degradation, with detailed discussions provided in the Appendix H. We also conduct experiments comparing KBE-DME with other multimodal dynamic evaluation methods, with detailed results provided in the Appendix I.

### 4.3. Ablation Study of Evaluation Method

We attempted to directly use strict character matching as an additional evaluation method to analyze the results. As shown in Figure 4, even after switching to a different evaluation method, the conclusions remain. In most cases, the difficulty control behaves as expected, and there is indeed a trend of diminishing marginal effects. Under strict string matching, the evaluation results of models become slightly lower. Overall, as the number of expansion hops increases, the difficulty does increase. However, when a model already performs poorly on the current question, generating an even more difficult question becomes challenging, which may lead to some fluctuations in accuracy. Consistent with previous conclusions, GPT-4o and Claude-3.5 continue to perform well, while LLaVA-OV remains the model with the largest accuracy fluctuations.

### 4.4. Ablation Study of Dynamic Generation Model

We apply the open-source model Qwen2-VL-72B instead of GPT-4o as Dynamic Generation Model on the subset of OK-VQA to conduct ablation experiments. We sample 380 instances after the filter process. We evaluated whether the model's responses were correct using string matching.

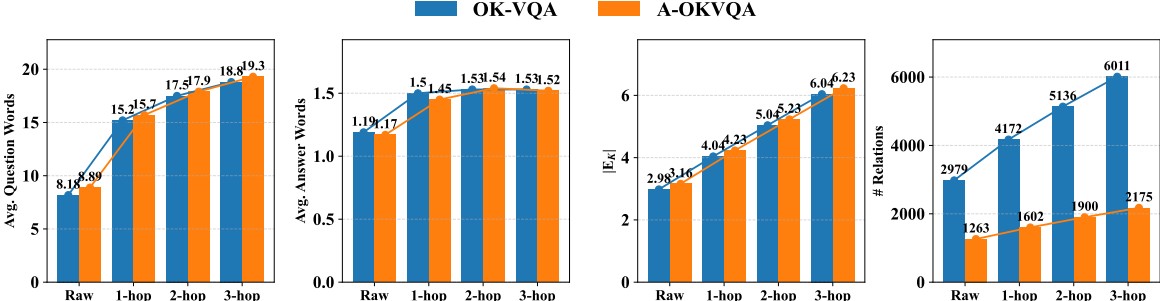

*Figure 6.* Several statistical metrics of original VQA data and the VQA questions generated with exploration of different hops. The statistical metrics including the average number of words in the questions, the average number of words in the answers, the average number of edges $|E_K|$ in the key subgraphs $G_K$, and the number of distinct relations among the triplets in the entire corresponding dataset.

The results are shown in the Figure 5. As shown, when using Qwen2-VL-72B to dynamically generate questions, the majority of the results still satisfy the requirements of difficulty control. The previous conclusions hold when applying different mllms as Dynamic Generation Model. What's more, even within the Qwen-VL family, Qwen-2.5-VL doesn't achieve much better performance on data generated by Qwen2-VL-72B than by GPT-4o. Therefore, for the dynamic data in our current framework, bias from Dynamic Generation Model does not affect the overall conclusion. We also experiment with using more diverse models as DGMs, details can be found in the Appendix G.

## 5. Quality Analysis

### 5.1. Statistics

To better analyze the differences between our newly generated data and the original VQA data, we conduct a statistical analysis on both sets of VQA samples. The results are presented in Figure 6. We observe that on both benchmarks, as the number of expansion hops increases, the newly generated VQA samples tend to have longer questions. This is often due to the fact that answering these questions requires longer reasoning chains, which in turn makes the questions more complex. We also find that the number of edges in the key subgraph increases steadily with more hops, and consequently, the number of relation types involved in the visual and textual knowledge triples also grows. These results demonstrate that the newly generated VQA questions achieve higher distributional diversity, greater question complexity, and longer rationales compared to the original or lower-hop expansions, thereby producing more challenging VQA problems. This suggests that through the KBE-DME framework, we can evolve a static VQA benchmark into a dynamically changing dataset with controllable difficulty for dynamic evaluation.

*Table 2.* Human_Study of KBE-DME on OK-VQA Benchmark. We manually assessed the average quality of generated VQA questions under different hops from three perspectives: Reasonable, Triplet_Correct and Alignment. The values in parentheses indicating inter-annotator agreement.

| | Reasonable | Triplets_Correct | Alignment |
|---|---|---|---|
| Average | 95.0 (90.1%) | 96.8 (93.6%) | 97.9 (95.7%) |

### 5.2. Human Study

In addition to ensuring the diversity of dynamically generated data with controllable difficulty, it is also essential to guarantee their quality. We sampled 150 dynamically generated evaluation VQA instances with different hops from the OK-VQA dataset for human study. We assessed the dynamic generation process of VQA questions from three perspectives: (1) whether the newly generated VQA sample is itself reasonable as a VQA problem (Reasonable); (2) whether each triplet in the corresponding set of key knowledge triples $K$ is correct (Triplet_Correct); and (3) whether the generated VQA question is consistent with its corresponding key knowledge triplets (Alignment). The evaluation results are presented in the Table 2. Prompt can be seen in Appendix J. The human evaluation results demonstrate that our dynamic evaluation framework, KBE-DME, can generate high-quality VQA data with correct and well-aligned key triplets. This further validates the accuracy of our framework and the reliability of the generated VQA data.

## 6. Conclusion

In this paper, we first introduce a graph-based representation to model VQA data and the dynamic evaluation process. We then propose KBE-DME, a dynamic multimodal evaluation framework. Building upon two static VQA benchmarks,

OK-VQA and A-OKVQA, we dynamically construct test samples with varied difficulty levels and conduct comprehensive analyses of the diversity and quality of our dynamically constructed data. We further apply dynamic evaluation on five different open and closed-source mllms. Experimental results show that KBE-DME can dynamically generate high-quality test data with controllable difficulty, while the evaluation results reveal consistent performance degradation of all tested models on harder data. Overall, KBE-DME provides a generalizable framework that can be applied to diverse multimodal benchmarks, effectively alleviates the risk of data saturation and contamination.

## Impact Statement

Our released dataset sources data from open-source datasets as indicated, following their license and copyright restrictions. Our released dataset containing synthetic data is for research only and does not aim at conveying any information about real-life.

## Acknowledgments

This work was supported by Beijing Natural Science Foundation (L253001), Key Laboratory of Science, Technology and Standard in Press Industry (Key Laboratory of Intelligent Press Media Technology) and National Engineering Research Center of New Electronic Publishing Technologies. We appreciate the anonymous reviewers for their helpful comments. Xiaojun Wan is the contact author.

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

| VQA $S_0$ | VQA $S_1$ | VQA $S_2$ | VQA $S_3$ |
|---|---|---|---|
| $I_0$ 
 | $I_0$ 
 | $I_0$ 
 | $I_0$ 
 |
| $Q_0$ 
 What kind of birds are those? | $Q_1$ What taxonomic order do the red and black birds in this image belong to? | $Q_2$ What is the typical habitat for the birds depicted in this image? | $Q_3$ What is the primary vegetation found in the typical habitat of the birds shown in this image? |
| $A_0$ woodpecker | $A_1$ piciformes | $A_2$ forests | $A_3$ trees |
| $K_0$ 

 V1: (IMAGE, depict, BIRDS) 
 V3: (BIRDS, have color, red and black) 
 T1: (WOODPECKER, type of, BIRD) | $K_1$ 

 V1: (IMAGE, depict, BIRDS) 
 V3: (BIRDS, have color, red and black) 
 T1: (WOODPECKER, type of, BIRD) 
 T5: (WOODPECKER, Taxonomic order, PICIFORMES) | $K_2$ 

 V1: (IMAGE, depict, BIRDS) 
 V3: (BIRDS, have color, red and black) 
 T1: (WOODPECKER, type of, BIRD) 
 T5: (WOODPECKER, Taxonomic order, PICIFORMES) 
 T6: (PICIFORMES, typical habitat, FORESTS) | $K_3$ 

 V1: (IMAGE, depict, BIRDS) 
 V3: (BIRDS, have color, red and black) 
 T1: (WOODPECKER, type of, BIRD) 
 T5: (WOODPECKER, Taxonomic order, PICIFORMES) 
 T6: (PICIFORMES, typical habitat, FORESTS) 
 T7: (FORESTS, primary vegetation, TREES) |

*Figure 7.* An example from our data construction process on OK-VQA. $S_0$ denotes the original VQA sample, where $I_0$, $Q_0$, and $A_0$ represent the corresponding image, question, and answer, and $K_0$ is the associated set of key knowledge triples. For the generated data $S_i$ at different hop levels i, we keep the input image unchanged while reconstructing the corresponding questions and answers by altering the composition of the key knowledge triplets.

## A. Case Study

We present an example of the data construction process from OK-VQA, as illustrated in the Figure 7. Our KBE-DME first analyzes the original VQA problem and identifies the key knowledge triples. After the filtering process and the possible Re-Selection step (as illustrated in the Figure 2), KBE-DME searches for textual knowledge triplets whose subjects correspond to the original answer (such as T5: (WOODPECKER, Taxonomic order, PICIFORMES)). Following an additional filtering step, a selected triplet is incorporated into the original key knowledge set, and based on this updated set of key triplets, KBE-DME generates a new VQA question as "What taxonomic order do the red and black birds in this image belong to?". By repeating this process, we can iteratively expand and generate new VQA question–answer pairs corresponding to different sets of key knowledge triplets, indicating the effectiveness of our proposed framework.

## B. Full Results

We list the full experimental results of our figures here. Figure 3 is corresponding to Table 3, Figure 4 is corresponding to Table 4, Figure 5 is corresponding to Table 5.

## C. Analysis of Output of Models

We calculated the average number of generated words for different models across different questions, and the results are shown in the Table 6. Although in some cases the average output length of the model does increase with task difficulty, it is not rigorous to analyze the model's effort in answering questions solely based on its average output length. There are two main reasons for this: (1) The answer lengths of different VQA questions are inconsistent, which leads to variability in the model's output length. For different dynamically generated questions, the number of words required in their corresponding answers may naturally vary. (2) The different models differ in settings and output style, which also leads to inconsistencies in output length. For example, the open-source LLaVA-OV model has not do RL training for reasoning capabilities, so it often outputs only the final answer without providing intermediate reasoning. In contrast, models such as Claude-3.5 not

*Table 3.* Main Results of five different MLLMs on original static benchmark(Raw) and our generated dynamic benchmark with exploration of different(1-3) hops through our KBE-DME framework.

| Model | OK-VQA | | | | A-OKVQA | | | |
|---|---|---|---|---|---|---|---|---|
| | Raw | 1-hop | 2-hop | 3-hop | Raw | 1-hop | 2-hop | 3-hop |
| GPT-4o | 50.33 | 47.62 | 42.07 | 39.19 | 60.13 | 50.82 | 41.34 | 36.27 |
| Gemini-2.5-pro | 49.94 | 41.55 | 36.87 | 32.92 | 58.99 | 46.57 | 34.48 | 32.03 |
| Claude-3.5 | 52.26 | 46.03 | 41.82 | 37.60 | 57.68 | 48.85 | 38.73 | 32.84 |
| LLaVA-OV | 53.46 | 36.53 | 31.45 | 29.52 | 60.78 | 38.56 | 30.56 | 25.33 |
| Qwen-2.5-VL | 47.35 | 42.24 | 37.68 | 33.04 | 57.35 | 43.95 | 37.42 | 30.88 |

*Table 4.* Main Results of five different MLLMs on original static benchmark(Raw) and our generated dynamic benchmark with exploration of different(1-3) hops through our KBE-DME framework with strict character matching as evaluation method.

| Model | OK-VQA | | | | A-OKVQA | | | |
|---|---|---|---|---|---|---|---|---|
| | Raw | 1-hop | 2-hop | 3-hop | Raw | 1-hop | 2-hop | 3-hop |
| GPT-4o | 47.00 | 32.77 | 29.17 | 23.91 | 53.27 | 35.13 | 25.33 | 26.31 |
| Gemini-2.5-pro | 46.38 | 30.29 | 25.57 | 22.59 | 53.43 | 32.84 | 25.49 | 24.18 |
| Claude-3.5 | 49.36 | 32.11 | 30.91 | 26.81 | 49.51 | 33.44 | 24.84 | 24.67 |
| LLaVA-OV | 53.19 | 23.17 | 19.85 | 18.14 | 58.82 | 25.49 | 18.30 | 15.85 |
| Qwen-2.5-VL | 41.24 | 24.41 | 24.26 | 21.08 | 47.06 | 28.10 | 21.08 | 18.63 |

only generate a certain amount of reasoning chains but also tend to follow specific output formats.

The two points above imply that it is not rigorous to assess the amount of effort a model spends on answering questions, whether by comparing the average output lengths of different models on questions of the same difficulty, or by comparing a single model's output lengths across questions of different difficulty levels. It is inherently difficult to measure a model's "effort" in answering questions. To address this, we are conducting a human evaluation experiment on a subset of OK-VQA to directly assess the difficulty of dynamic generated VQAs.

## D. Human Study of Difficulty Control

We randomly sampled 150 VQA questions from the Dynamic VQAs from OK-VQA dataset for human difficulty annotation. The results show that the agreement between human annotators and our difficulty control is 84.15%, while the inter-annotator agreement is 64.06% (measured using the Pearson-r correlation coefficient). The annotation prompt is shown in the Figure 8.

## E. Experimental Details

**Evaluated MLLMs**   Our evaluation covers both closed-source models, including GPT-4o(OpenAI et al., 2024), Gemini-2.5-pro(Comanici et al., 2025), and Claude(Anthropic, 2024), as well as open-source models und, namely LLaVA-OV-7B(Li et al., 2024b) and Qwen-2.5-VL-7B(Bai et al., 2025). To ensure fair comparison in answering VQA questions, we restrict the length of the models' responses. Considering possible alias cases, we employ GPT-4o to determine whether the response of the tested model to a VQA question corresponds to the provided answer.

**Computational Cost**   To generate three dynamic VQA samples from one static VQA sample, we need to apply the MLLM approximately 11 times, an average of 3.67 model calls per newly generated question. The cost is significantly lower and the construction efficiency is much higher compared with manually reconstructing the dataset with different difficulty.

*Table 5.* Main Results of five different MLLMs on original static benchmark(raw) and our generated dynamic benchmark with exploration of different(1-3) hops through our KBE-DME framework with Qwen2-VL-72B as Dynamic Generation Model. We apply strict character matching as the evaluation method.

| Model | OK-VQA | | | |
|---|---|---|---|---|
| | Raw | 1-hop | 2-hop | 3-hop |
| GPT-4o | 50.00 | 29.47 | 28.16 | 25.53 |
| Gemini-2.5-pro | 45.26 | 26.58 | 22.89 | 23.42 |
| Claude-3.5 | 46.84 | 30.00 | 29.47 | 25.00 |
| LLaVA-OV | 52.11 | 20.00 | 20.53 | 18.16 |
| Qwen-2.5-VL | 42.89 | 27.11 | 23.68 | 22.37 |

*Table 6.* Average words number of output of five different MLLMs on original static benchmark(raw) and our generated dynamic benchmark with exploration of different(1-3) hops through our KBE-DME framework.

| Model | OK-VQA | | | | A-OKVQA | | | |
|---|---|---|---|---|---|---|---|---|
| | Raw | 1-hop | 2-hop | 3-hop | Raw | 1-hop | 2-hop | 3-hop |
| GPT-4o | 4.14 | 4.15 | 4.52 | 4.62 | 3.76 | 3.62 | 4.32 | 4.73 |
| Gemini-2.5-pro | 3.26 | 2.70 | 2.63 | 2.79 | 2.65 | 2.63 | 2.83 | 2.77 |
| Claude-3.5 | 8.78 | 9.00 | 9.06 | 9.15 | 8.98 | 9.20 | 9.23 | 9.12 |
| LLaVA-OV | 1.22 | 1.24 | 1.23 | 1.23 | 1.20 | 1.20 | 1.83 | 1.25 |
| Qwen-2.5-VL | 6.99 | 7.26 | 7.16 | 7.14 | 7.41 | 7.26 | 7.19 | 7.35 |

## F. Generalization on Other VQA Datasets

We randomly sampled a 1K subset from VQA-v2(Goyal et al., 2017) for evaluation. The results are shown in Table 7. Additionally, as Claude-3.5 is no longer available, we report results using Claude-4.5(Anthropic, 2025). These results demonstrate that our framework generalizes well to VQA-v2. In particular, most models exhibit consistent performance degradation as the hop number increases, indicating that our difficulty control mechanism remains effective beyond the two datasets on main text. Moreover, LLaVA-OV shows the largest drop from Raw to 1-hop, which is consistent with our findings.

*Table 7.* Model performance on dynamic evaluation with KBE-DME framework on subset of VQA-v2.

| Model | Raw | 1-hop | 2-hop | 3-hop |
|---|---|---|---|---|
| GPT-4o | 58.68 | 33.74 | 30.07 | 22.98 |
| Gemini-2.5-pro | 73.27 | 29.83 | 27.65 | 22.25 |
| Claude-4.5 | 73.11 | 40.34 | 39.36 | 33.01 |
| LLaVA-OV | 71.88 | 23.96 | 21.27 | 15.65 |
| Qwen-2.5-VL | 59.41 | 26.16 | 26.65 | 20.05 |

## G. Extra Experiments about Different DGMs

We add experiment using Gemini 2.5 Pro as DGM. We test models on raw and 1-hop samples, results are shown in Table 8. Since different DGMs yield different filtered data, we also report raw performance in parentheses. Due to the unavailability of the Claude 3.5 API, we evaluate Claude 4.5 under Gemini-based DGM, other corresponding data in the Table 8 are taken from Table 4 and 5. As shown in table, although model performance varies across different DGMs, the relative change with respect to the raw setting remains stable overall. We observe that, performance consistently decreases on the 1-hop data

---

**Difficulty Human Study Prompt**

We will provide two Visual Question Answering instances. Please evaluate which of the two VQA questions is more difficult. If the first VQA question is more difficult than the second, output 1; otherwise, output 2.
======================VQA1======================
Image: The given image.
Question: {Question1}
Answer: {Answer1}.
======================VQA2======================
Image: The given image.
Question: {Question2}
Answer: {Answer2}.
Evaluation Output:

---

*Figure 8.* Difficulty Human Study Prompt.

across all models and LLaVA-OneVision still exhibits the largest performance drop. This demonstrates that our dynamic evaluation framework consistently mitigates data contamination and saturation issues under different settings, while yielding stable and consistent conclusions, which can not be achieved by traditional static evaluation.

*Table 8.* Extra experiments about different DGMs. We show the performance of models on raw and 1-hop samples under different DGMs.

| DGM | Tested Model | | | | |
|---|---|---|---|---|---|
| | GPT-4o | Gemini-2.5-pro | Qwen-2.5-VL | Claude-3.5/4.5 | LLaVA-OV |
| GPT-4o | 32.77(47.00) | 30.29(46.38) | 24.41(41.24) | 32.11(49.36) | 23.17(53.19) |
| Gemini-2.5-pro | 35.35(53.91) | 37.11(52.15) | 25.78(46.29) | 38.87(51.17) | 25.78(55.47) |
| Qwen-2-VL-72B | 29.47(50.00) | 26.58(45.26) | 27.11(42.89) | 30.00(46.84) | 20.00(52.11) |

## H. Preliminary Analysis of Performance Degradation

To further investigate the factors contributing to performance degradation, we conduct a preliminary analysis. We define difficulty in a relative sense in main text, where the 1-hop expanded dataset is overall more challenging than the original dataset. Here, for preliminary analysis, we additionally try to use hop count as a proxy for absolute difficulty to enable a more fine-grained comparison. Specifically, we stratify both the original and 1-hop datasets based on the number of edges in the corresponding key subgraphs. We select samples from the original raw dataset with 3-edge key subgraphs as the control group. From the 1-hop dataset, we select two groups: 1hopEZ with 3-edge key subgraphs, and 1hopHard with 4-edge key subgraphs. To ensure a fair comparison, we randomly sample nearly 1k instances for each group to keep the dataset sizes consistent. We then evaluate model accuracy across these three groups, as shown in Table 9.

We observe that although the original set and 1hopEZ have the key subgraph with identical edge counts, model accuracy drops substantially on raw to 1hopEZ, particularly for the open-source model LLaVA-OV. In contrast, when comparing 1hopEZ and 1hopHard, both of which are dynamically generated, performance further decreases on 1hopHard. However, the magnitude of this drop is notably smaller than the drop observed from the original set to 1hopEZ. This suggests that, for certain models, data contamination may pose a significant risk.

## I. Comparison of VLB

We reproduce one of Text Bootstrapping settings of VLB on OK-VQA, specifically the **L4** which is Adding Irrelevant Context (Hard) strategy. The results are shown in Table 10. For Claude-4.5, results are newly evaluated, while the others are taken from our reported results under the same evaluation setting in Table 4. The average question length of L4 is 97.2 words. From these results, we make the following observations: 1. Limited difficulty increase. Although L4 is designed as a hard setting, model performance only slightly decreases compared to Raw, and remains consistently higher than that of our

*Table 9.* Performance of MLLMs on raw, 1hopEZ and 1hopHard subsets. The number in parentheses denotes the number of edges in the corresponding key subgraph.

| Model | Raw(3) | 1hopEZ(3) | 1hopHard(4) |
|-------|--------|-----------|-------------|
| GPT-4o | 46.13 | 33.85 | 32.30 |
| Gemini-2.5-pro | 46.02 | 31.42 | 29.31 |
| Claude-3.5 | 49.12 | 33.19 | 30.53 |
| Qwen-2.5-VL | 42.04 | 26.00 | 23.01 |
| LLaVA-OV | 54.09 | 24.45 | 21.90 |

*Table 10.* Comparison between VLB and KBE-DME. We reproduce **L4** method of VLB on OK-VQA.

| Model | Raw | L4 | 1-hop | 2-hop | 3-hop |
|-------|-----|-----|-------|-------|-------|
| GPT-4o | 47.00 | 39.69 | 32.77 | 29.17 | 23.91 |
| Gemini-2.5-pro | 46.38 | 44.76 | 30.29 | 25.57 | 22.59 |
| Claude-4.5 | 56.75 | 55.83 | 36.07 | 32.31 | 29.48 |
| LLaVA-OV | 53.19 | 50.02 | 23.17 | 19.85 | 18.14 |
| Qwen-2.5-VL | 41.24 | 32.69 | 24.41 | 24.26 | 21.08 |

dynamically generated 1-hop questions. This suggests that L4 provides a weaker form of difficulty control. 2. Inconsistent effect across models. The performance drop introduced by L4 varies substantially across models, indicating that its impact is less stable. Moreover, since L4 preserves the original answer, the task may rely on recovering and answering the underlying original question, rather than directly addressing the newly constructed question. 3. Significant distribution shift. L4 substantially increases question length (97.2 vs. 8.18 words in Raw), whereas our method maintains a moderate increase (up to 18.8 in 3-hop). This suggests that L4 alters the distribution more drastically, while our approach preserves more natural question forms. Overall, these results suggest that our KBE-DME achieves more precise and consistent difficulty control compared to L4 method of VLB.

## J. Prompt

The prompts are presented as follows:

**Graph Extraction Prompt**

You are a helpful assistant. You need to analyze the visual information subgraph and textual information subgraph implied in the input information of a VQA instance, which includes an image, a question, and an answer. Please output and number the visual and textual knowledge subgraphs in the form of knowledge triples.
Here is an example of output visual and textual knowledge subgraphs.
Visual Information Subgraph:
V1.(Image, contains, motorcycle)
V2.(motorcycle, has color, black)
V3.(motorcycle, has component, engine)
V4.(motorcycle, has feature, two wheels)
V5.(motorcycle, is on, road or track)
Textual Information Subgraph:
T1.(motorcycle, can be used for, race)
T2.(sport, has type, race)
T3.(race, requires, high speed vehicle)
T4.(high speed vehicle, includes, motorcycle)
Now, please generate the visual and textual information subgraphs for a VQA example.
Here is the VQA instance:
Image: The given image
Question: {VQA Question}
Answer: {VQA Answer}
Please generate: the visual knowledge subgraph implied in the image
and the textual knowledge subgraph implied in the question and answer.
Visual Information Subgraph:
Textual Information Subgraph:

*Figure 9.* Graph extraction prompt.

**Key Triplets Extraction Prompt**

You are a helpful assistant. You will be given a (VQA) Visual Question Answering instance that includes an input image, a question, and its corresponding answer, as well as a set of corresponding visual and textual information in the form of triplets. The triplets in the visual information contain only visual information implied in the image, excluding any textual background knowledge. The triplets in the textual information include only relevant textual background knowledge. Please select the necessary key information triplets that are required to answer this VQA question. Below are some examples:

======Example1======:
VQA Question: The man wearing a hat what is the name of that hat?
VQA Answer: cowboy hat
Visual Information triplets:
V1: (IMAGE, depict, MAN)
V2: (MAN, wear, HAT)
V3: (HAT, have type, COWBOY HAT)
V4: (MAN, ride, HORSE)
V5: (HORSE, is on, PATH)
V6: (PATH, is in, MOUNTAINOUS AREA)
V7: (IMAGE, contain, BACKPACK)
V8: (BACKPACK, have color, red)
Textual Information triplets:
T1: (COWBOY HAT, is a type of, HAT)
T2: (COWBOY HAT, typically worn by, COWBOYS)
T3: (COWBOY, commonly associated with, HORSE RIDING)
T4: (COWBOY HAT, used for, SUN PROTECTION)
Key information triplets to answer the question:
V1: (IMAGE, depict, MAN)
V2: (MAN, wear, HAT)
V3: (HAT, have type, COWBOY HAT)
======Example2======:
VQA Question: How many teeth does this animal use to have?
VQA Answer: 26
Visual Information triplets:
V1: (IMAGE, depict, CAT)
V2: (CAT, have color, beige)
V3: (CAT, is on, WINDOWSILL)
V4: (WINDOWSILL, is part of, WINDOW)
V5: (CAT, is in state, RELAXED)
Textual Information triplets:
T1: (ANIMAL, typically have, TEETH)
T2: (CAT, category of, ANIMAL)
T3: (CAT, usually have, 26 TEETH)
Key information triplets to answer the question:
V1: (IMAGE, depict, CAT)
T3: (CAT, usually have, 26 TEETH)
Now, please select the Key information triplets given the image, question and answer of a VQA example with its corresponding Visual and Textual Information triplets.
======VQA Input======
Image: The given image
VQA Question: {VQA_Q}
VQA Answer: {VQA_A}
Visual Information triplets:
{Visual_Information_triplets_str}
Textual Information triplets:
{Textual_Information_triplets_str}
Key information triplets to answer the question:

17

*Figure 10.* Key triplets extraction prompt.

---

**Knowledge Generation Prompt**

You are a helpful assistant. You will receive a VQA example. Please generate some knowledge triplets related to the answer. You should understand the meaning of the answer by combining the image and the question in the VQA, and then generate answer-related knowledge triplets. The newly generated knowledge triplets should not conflict with the information in the original VQA example. A triplet can be represented as (s, r, o), where s is the subject (an object), r is the corresponding relation, and o is either an attribute of the object or another object. Use uppercase for objects and lowercase for attributes. Here, s is the relation subject, r is the related relation, and o is the result of the relation corresponding to s. Generated knowledge triplets (s, r, o) should follow the following requirements:

1. The subject (s) must always be the answer itself.

2. The relation (r) must be specific and unique. Do not use vague terms like is a or has; instead, refine them into clear categories or attributes, such as taxonomic_class, primary_covering, foot_type.

3. Please ensure that within a triplet (s, r, o), the object (o) is unique given the specified subject (s) and relation (r). In a triplet (s, r, o), the o must be an object, not an attribute.

4. The output format must strictly be one triplet per line: (s, r, o).

Below are some examples:

=====================Example1=====================
VQA_Question: What country does this appear to be?
VQA_Answer: scotland
Answer Related Knowledge Triples:
(SCOTLAND, geographic_location, united_kingdom)
(SCOTLAND, primary_landscape, highlands)
(SCOTLAND, common_tree_type, deciduous)
=====================Example2=====================
VQA_Question: What animal is this boat mimicing?
VQA_Answer: duck
Answer Related Knowledge Triples:
(DUCK, taxonomic_class, AVES)
(DUCK, taxonomic_order, ANSERIFORMES)
(DUCK, taxonomic_family, ANATIDAE)
(DUCK, common_category, WATERFOWL)
(DUCK, typical_habitat, WATER)
(DUCK, primary_covering, FEATHERS)
(DUCK, mouth_structure, BEAK)
(DUCK, foot_type, WEBBED_FEET)
(DUCK, typical_sound, QUACK)
(DUCK, diet_type, OMNIVORE)
Below is the VQA sample for generating extended knowledge:
Image: The given image
VQA_Question: {VQA_Q}
VQA_Answer: {VQA_A}
After generating the relevant (s, r, o) triplets, check each triplet individually and generate all possible o values for the given s and r. If a tuple (s, r, o) contains multiple outputs for the same s and r, delete that tuple. If o is an attribute rather than an object, also delete that tuple.
Answer Related Knowledge Triples:

---

*Figure 11.* Knowledge generation prompt.

---

**Representative Filter Prompt**

Here are some triplets related to a VQA example. Each triplet is composed of (s, r, o). I will provide you with the corresponding VQA example, and based on the VQA context, you need to determine whether the given o is representative for the specified s and r.
If it is representative, please output Yes.
If the relation is too broad or ambiguous to determine a unique representative, simply output No.
Please output only the triplet numbers and their corresponding results: Yes or No.
You must evaluate every triplet and output the corresponding result in order.
Here are some examples:
==========Example1==========
VQA_Question: What type of platform should this vehicle be on?
VQA_Answer: track
Related Triplets:
1.(TRACK, primary_use, TRANSPORTATION)
2.(TRACK, common_association, TRAINS)
3.(TRACK, typical_material, STEEL)
Representative Judgment:
1.No
2.Yes
3.Yes
==========Example2==========
VQA_Question: Name the material used to make this car seat shown in this picture?
VQA_Answer: cloth
Related Triplets:
1.(CLOTH, typical_use, UPHOLSTERY)
2.(CLOTH, material_origin, TEXTILE)
3.(CLOTH, common_source, PLANT_FIBERS)
Representative Judgment:
1.Yes
2.No
3.Yes
The following are examples to be judged:
Image: The given image.
VQA_Question: {VQA_Q}
VQA_Answer: {VQA_A}
Related Triplets: {Related_Triplets}
Representative Judgment:

*Figure 12.* Representative filter prompt.

---

**Question Generation Prompt**

Please generate a new VQA question based on an original VQA question and the related information triplets. A triplet can be represented as (s, r, o), where s is the subject (an object), r is the corresponding relation, and o is either an attribute of the object or another object. Use uppercase for objects and lowercase for attributes. We will provide you with the triplets necessary for forming the new question. These triplets consist of those used to answer the original VQA question as well as additional newly introduced triplets. We will also specify the answer for the new question along with the corresponding answer information triplet. You should combine the given original question and all the knowledge triplets to generate the new VQA question. The new VQA question must ensure that its answer is the specified one and that it is related to the provided answer triplet. The new question must not contain the original question's answer, and it should require the use of all provided knowledge triplets in order to be answered. Apart from the information in the newly added triplets, the new question must not include more information than the original question. Below are some examples:
======Example======:
Original VQA Question: What country does this appear to be?
Original VQA Answer: scotland.
Related Information Triplets for Original VQA sample:
visual_triplets_list:
V2: (IMAGE, depict, SHEEP)
V3: (IMAGE, depict, LAND ROVER)
textual_triplets_list:
T1: (SHEEP, commonly found in, SCOTLAND)
T2: (LAND ROVER, associated with, BRITISH COUNTRYSIDE)
T3: (BRITISH COUNTRYSIDE, includes, SCOTLAND)
Related Information triplet for New Answer in Generated VQA sample:
(SCOTLAND, traditional_clothing, KILT)
New Answer in Generated VQA sample: KILT
Generated new VQA Question:
What is the traditional clothing of the country shown in this image?
Please generate a new VQA question based on the information provided below, following the given example and requirements. The provided information is as follows:
Original VQA Image: The given image.
Original VQA Question: {Ori_VQA_Q}
Original VQA Answer: {Ori_VQA_A}.
Related Information Triplets for Original VQA sample:
{Key_Triplets_str}
Related Information triplet for New Answer in Generated VQA sample:
{New_VQA_related_Triplets}
New Answer in Generated VQA sample:
{New_VQA_Answer}
Generated new VQA Question:

---

*Figure 13.* Question generation prompt.

Judging Prompt

Please analyze whether a given response to a VQA question matches its corresponding answer. If they match, output "Yes"; otherwise, output "No". Only output the judgment result "Yes" or "No". We will provide relevant image information to assist in the judgment.
Image: The given image.
Response: {Response}
Answer: {Answer}

*Figure 14.* Judging prompt.

