# OpenReview forum: "Dynamic Multimodal Evaluation via Knowledge-Enhanced Benchmark Evolution"
_ICML.cc/2026/Conference — ICML 2026 regular_

### Official Review · Reviewer_fFwX · 2026-03-07

**Soundness:** 3
**Presentation:** 3
**Significance:** 3
**Originality:** 3
**Overall Recommendation:** 4
**Confidence:** 3

**Summary:**

This paper proposes a dynamic multimodal evaluation framework, KBE-DME, which converts static VQA benchmark datasets into dynamic test sets with controllable difficulty. The framework explicitly models multimodal knowledge as a triple graph and regenerates questions through reselection and exploration strategies. The authors evaluate five representative multimodal large language models (MLLMs) on the OK-VQA and A-OKVQA datasets. Experimental results demonstrate that KBE-DME enables high-quality and difficulty-controllable dynamic evaluation for multimodal large language models.

**Compliance With Llm Reviewing Policy:**

Affirmed.

**Final Justification:**

My concerns have been largely addressed. Therefore, I will maintain my weak accept score.

**Key Questions For Authors:**

The key issues remain the same as above.

**Limitations:**

The authors should provide additional discussion on the limitations of their study and its potential societal impacts.

**Strengths And Weaknesses:**

Strengths of the paper include:
1. The paper introduces a clear graph-based formulation of VQA instances, enabling principled edits to the reasoning core rather than relying on superficial perturbations.
2. It proposes two general subgraph transformation operations, which provide fine-grained difficulty control through hops and measurable graph properties.
3. The framework evaluates five representative MLLMs on two widely used knowledge-intensive VQA datasets, OK-VQA and A-OKVQA, and the results provide some evidence of the method’s effectiveness.
4. The framework is well motivated and technically simple, and it partially addresses two key issues in current multimodal evaluation: data contamination and benchmark saturation.

Weaknesses of the paper include:
1. The core steps of the KBE-DME framework rely on GPT-4o as the dynamic generation and judgment model, while GPT-4o itself is also one of the five MLLMs being evaluated. This design raises concerns about circular evaluation and model bias. Although the authors conduct an ablation study replacing GPT-4o with Qwen2-VL-72B to test robustness to different generation models, the scope and strength of this experiment are limited, and it does not adequately resolve the core issue.
2. The cycle-checking mechanism has limitations: cycles naturally exist in legitimate knowledge graphs, so the rule may incorrectly remove valid reasoning structures. Additionally, KBE-DME requires generated VQA answers to be nouns, which seems inconsistent with real-world VQA scenarios and may lead to the loss of many valid samples.
3. The paper lacks direct empirical comparisons with other advanced dynamic evaluation methods or datasets. Table 1 only provides a functional comparison, without supporting experimental results, which weakens the persuasiveness of the framework’s competitiveness and advantages.
4. The claim of mitigating data contamination is only supported by indirect evidence (e.g., performance drops in first-hop questions), and the paper does not provide concrete contamination analysis.

---

> ### Author Rebuttal · Authors · 2026-03-31
>
> Thanks for your careful and valuable comments. We will address your concerns point by point.
>
> ### R to W1
> To further examine potential **circular evaluation and model bias**, **we conduct additional experiments using Gemini 2.5 Pro as the DGM** (Due to the space limitation, details can be seen in **our response to Reviewer zvvE, R to L1**). We compare model performance **when using GPT-4o, Gemini 2.5 Pro, and Qwen2-VL-72B as DGMs**, evaluating both raw and 1-hop generated questions.
>
> The results show that although model performance varies across different DGMs, the **relative change with respect to the raw setting remains stable**.
>
> We **consistently** observe that:
>
> (1) model performance decreases on 1-hop data.
>
> (2) LLaVA-OneVision exhibits the largest performance drop.
>
> **Overall, the conclusions remain consistent with those reported in the paper.**
>
> ### R to W2
> We primarily avoid **cycles** during dynamic generation to prevent infinite loops. While cyclic structures do occur, **their frequency is relatively low in our dataset.** For example, on OK-VQA, we observe **5**, **278**, and **449** cyclic cases across the **three expansion stages**, respectively.
>
> Notably, **cycles are rare in early expansions and become more frequent in later stages**. Although some cyclic reasoning paths may be valid, our framework only requires selecting a **single accurate and reasonable triplet** at each step. Therefore, we simply avoid cyclic triplets and instead select alternative valid ones, as long as they are correct.
>
> While this design may sacrifice some diversity, we prioritize **framework correctness**, especially given the relatively low occurrence of cycles.
>
>
>
> KBE-DME constrains answers to be **nouns** to **facilitate subsequent multi-hop expansion**. If the **expansion reaches the final step, this constraint can be relaxed, allowing answers beyond nouns**. Therefore, our framework can still generate VQA questions with non-noun answers.
>
> During the three expansion stages, **we filter out 406, 295, and 217 samples**, respectively, due to non-noun answers. Additionally, for raw VQA data with non-noun answers, we leverage the **Re-Selection** step to reconstruct valid samples: **among 1,043 filtered cases, 994 are successfully** converted into valid examples which meets the noun requirements.
>
> Overall, these constraints are introduced to improve the **accuracy and reliability** of the generated questions **with acceptable cost of samples**.
>
>
> ### R to W3
> In Table 1, NPHardEval, DyVal, and MPA do not support multimodal settings, and are therefore not directly comparable. As for VLB, we are currently conducting additional experiments; if time permits, we will report VLB results in the second rebuttal phase.
>
>
> ###  R to W4
> Regarding precise data contamination analysis, **existing contamination methods often introduce changes in difficulty of questions to models**. As a result, directly applying such methods and evaluating performance across different difficulty levels **would still confound the effects of contamination and difficulty**. Therefore, before identifying a contamination method that does not alter difficulty, we conduct a more **controlled preliminary analysis of performance degradation**.
>
> Specifically, (due to space limitation, details can be seen in our response to Reviewer pXoY R to Q3), we compare settings with controlled difficulty. For original questions and 1-hop generated questions of **comparable difficulty**, model accuracy drops substantially. In contrast, when comparing **1-hopEZ** and **1-hopHard**, although performance further decreases on 1-hopHard, **the magnitude of this drop is notably smaller than that from raw to 1-hopEZ.**
>
> Combined with results from longer-hop expansions, these findings suggest that the performance drop from raw to 1-hop is **not primarily driven by increased difficulty**, but is more likely attributable to **potential data contamination effects**.
>
> ### R to L
> Thank you for the valuable suggestion. We will incorporate discussions on limitations and potential societal impacts into the revised version of the paper.

---

> > ### Author Rebuttal · Reviewer_fFwX · 2026-04-04
> >
> > The rebuttal addressed some of my concerns. However, the authors' claim that they introduced constraints considering the relatively low frequency of cycles did not fully convince me. Additionally, I look forward to seeing the authors' report on the VLB results. Therefore, I have decided to maintain my score for now.

---

> > > ### Author Response · Authors · 2026-04-06
> > >
> > > We thank the reviewer for the constructive feedback. Below we address the remaining concerns and provide additional clarifications and results.
> > >
> > >
> > >
> > > ### R to W2(cycles)
> > >
> > > Thank you for the insightful comment. We agree that our previous justification based only on cycle frequency was insufficient.
> > >
> > > We will elaborate on this design from the following aspects.
> > >
> > > **First, we clarify that cycles are preserved in the original data.** Cyclic structures naturally exist in the graph, and we do **not remove or alter them**. The constraint is only applied during *dynamic exploration*(as shown in right of line263), where we avoid selecting triplets that would introduce cycles.
> > >
> > > **Second, cycles are rare in the key subgraphs.** While 575 raw samples contain cycles in the full graph, only 42 retain cycles in their key subgraphs, which are the actual targets of dynamic exploration.
> > >
> > > **Third, and most importantly, our extra experiments show that introducing cycles during expansion negatively affects difficulty control.** Using 449 cycle-filtered samples from the 2-hop -> 3-hop stage, we conduct an ablation by allowing cyclic explorations. As shown below, **allowing cycles consistently leads to higher accuracy on 3-hop questions than 2-hop ones across all models**, indicating that the generated 3-hop questions become easier rather than more difficult.
> > >
> > > | Model          | 2hop  | 3hop (w/cycles) |
> > > | -------------- | ----- | --------------- |
> > > | GPT-4o         | 25.39 | 46.55           |
> > > | Gemini-2.5-pro | 28.09 | 47.85           |
> > > | Claude-4.5     | 26.28 | 49.22           |
> > > | Qwen-2.5-VL    | 20.94 | 41.20           |
> > > | LLaVA-OV       | 18.71 | 42.76           |
> > >
> > > An extra human study further confirms that **50% of the 3-hop(w/cycles) questions are easier than the original 2-hop ones**, whereas our method obtains an 84.15% probability of increasing difficulty (**appendix D**).
> > >
> > > Additionally, we examine the 42 raw samples whose key subgraphs contain cycles. In these samples, GPT-4o shows a clear decrease in accuracy from Raw to multi-hop settings (52.38 ->40.48->16.67->11.90), indicating that the **relative difficulty control remains effective even when cycles exist in the raw data**.
> > >
> > > These results suggest that cycles themselves are not inherently problematic, as they naturally exist in some raw data. The issue arises during **dynamic exploration**, where allowing cycle-inducing explorations may introduce degenerate reasoning paths(e.g., revisiting previously explored nodes without adding new information). This can lead to simpler reasoning chains, thereby weakening the intended difficulty progression.
> > >
> > > Since dynamic expansion only requires selecting any valid triplet, we therefore avoid cycle-inducing explorations to ensure the stability of difficulty control.
> > >
> > >
> > > ### R to VLB results
> > >
> > > Due to the lack of fully released code for the Image Bootstrapping in VLB, we reproduce its **Text Bootstrapping setting on OK-VQA**, specifically the **L4 *Adding Irrelevant Context (Hard)*** strategy. The results are shown below. For Claude-4.5, results are newly evaluated, **while the others are taken from our reported results under the same evaluation setting (Table 4).**
> > >
> > > | Model          | Raw   | L4    | 1Hop  | 2Hop  | 3Hop  |
> > > | -------------- | ----- | ----- | ----- | ----- | ----- |
> > > | GPT-4o         | 47.00 | 39.69 | 32.77 | 29.17 | 23.91 |
> > > | Gemini-2.5-pro | 46.38 | 44.76 | 30.29 | 25.57 | 22.59 |
> > > | Claude-4.5     | 56.75 | 55.83 | 36.07 | 32.31 | 29.48 |
> > > | LLaVA-OV       | 53.19 | 50.02 | 23.17 | 19.85 | 18.14 |
> > > | Qwen-2.5-VL    | 41.24 | 32.69 | 24.41 | 24.26 | 21.08 |
> > >
> > > We further analyze the average question length:
> > >
> > > | Raw  | L4   | 1Hop | 2Hop | 3Hop |
> > > | ---- | ---- | ---- | ---- | ---- |
> > > | 8.18 | 97.2 | 15.2 | 17.5 | 18.8 |
> > >
> > > From these results, we make the following observations:
> > >
> > > **(1) Limited difficulty increase.** Although L4 is designed as a ***hard*** setting, model performance only slightly decreases compared to Raw, and **remains consistently higher than that of our dynamically generated 1-hop questions.** This suggests that L4 provides a weaker form of difficulty control.
> > >
> > > **(2) Inconsistent effect across models.** The performance drop introduced by L4 **varies substantially across models**, indicating that its impact is less stable. Moreover, since L4 preserves the original answer, the task may **rely on recovering and answering the underlying original question**, rather than directly addressing the newly constructed question.
> > >
> > > **(3) Significant distribution shift.** L4 substantially increases question length (97.2 vs. 8.18 words in Raw), whereas our method maintains a moderate increase (up to 18.8 in 3-hop). This suggests that L4 alters the distribution more drastically, **while our approach preserves more natural question forms.**
> > >
> > > Overall, these results suggest that our KBE-DME achieves more precise and consistent difficulty control compared to L4.
> > >
> > > **We will add these discussion in the revised version.**

---

### Official Review · Reviewer_uYYr · 2026-03-12

**Soundness:** 3
**Presentation:** 3
**Significance:** 3
**Originality:** 3
**Overall Recommendation:** 4
**Confidence:** 4

**Summary:**

This paper proposes KBE-DME, a framework for dynamic multimodal evaluation aimed at addressing benchmark saturation and potential data contamination in current MLLM assessment. The core idea is to represent a VQA instance as a multimodal knowledge graph composed of visual and textual knowledge triplets, and then evolve this graph to generate new evaluation samples. The framework uses two main operations: Re-selection, which shifts attention to different key visual information within the image, and Exploration, which introduces external knowledge triplets to expand the reasoning chain. By controlling the number of graph hops, the method aims to generate test questions with adjustable difficulty. Experiments on OK-VQA and A-OKVQA with several representative MLLMs suggest that the framework can produce dynamic benchmarks of varying difficulty and induce performance degradation as the hop count increases.

**Compliance With Llm Reviewing Policy:**

Affirmed.

**Key Questions For Authors:**

1. Can the authors provide more direct evidence that the observed degradation on dynamically evolved benchmarks reflects reduced benchmark contamination, rather than confounding factors such as increased linguistic complexity, generator noise, or answer-format mismatch?
2. Can the authors quantify the error rates of different stages of the generation pipeline (e.g., triplet extraction, filtering, external knowledge expansion, and final QA synthesis), ideally with a breakdown by hop count?
3. As the generated questions become more knowledge-intensive and potentially more open-ended, how robust are the evaluation metrics to paraphrases or semantically equivalent answers? Could metric sensitivity partially account for the observed accuracy drop?

**Limitations:**

The framework is currently validated only on knowledge-centric VQA benchmarks, so its broader applicability remains uncertain. In addition, because the benchmark-generation process depends heavily on MLLMs, the resulting data quality is vulnerable to hallucinations and other generation errors. Most importantly, although the paper is motivated by contamination concerns, the current experiments do not yet cleanly isolate contamination reduction from other explanations for the observed increase in benchmark difficulty. These limitations should be discussed more explicitly in the paper.

**Strengths And Weaknesses:**

Strengths:
1. The formulation of VQA instances as multimodal knowledge-triplet graphs is a useful abstraction. It provides a structured way to evolve benchmark items beyond surface-level perturbation and is a meaningful step toward more dynamic evaluation.
2. Benchmark saturation and potential contamination are real concerns in multimodal evaluation. The paper tackles a timely and relevant problem, and the proposed dynamic-evaluation paradigm is potentially valuable for keeping evaluation suites challenging over time.
3. The hop-based graph expansion mechanism is intuitive and practically useful. The experiments provide evidence that increasing hop count makes the generated benchmark more challenging, which is one of the paper’s more convincing empirical findings.
Weaknesses:
1. The paper shows that dynamically generated questions are harder, but it does not fully isolate whether the observed performance drop is due to reduced contamination, as opposed to other confounding factors such as increased linguistic complexity, generation noise, answer-format mismatch, or reduced naturalness of the generated questions.
2. Since the framework depends on MLLMs to extract triplets, expand external knowledge, and synthesize new QA pairs, the final benchmark quality is sensitive to generator errors. The paper includes some quality validation, but it would benefit from a more granular analysis of error sources across pipeline stages and hop counts.
3. The empirical study focuses on knowledge-intensive VQA benchmarks. This is a reasonable starting point, but it leaves open how broadly the framework generalizes even across other multimodal evaluation settings with different output structures or supervision formats.
4. While the paper and appendix provide useful details, reproducibility would still benefit from a clearer presentation of the full prompting / filtering / thresholding pipeline and its sensitivity to design choices.

---

> ### Author Rebuttal · Authors · 2026-03-31
>
> Thanks for your careful and valuable comments. We will address your concerns point by point.
>
> ###  R to W1&Q1&L1
> We conduct a **more fine-grained analysis of the performance drop by isolating the effect of difficulty** (Due to space limitations, details can be seen in our **response to Reviewer pXoY, R to Q3**). The results show that, even when the **original questions** and **1hopEZ** have comparable difficulty, model accuracy drops substantially, particularly for the open-source model LLaVA-OneVision.
>
> In contrast, when comparing **1hopEZ** and **1hopHard**, both generated via one-step expansion, performance further decreases on 1hopHard, but the **magnitude of this drop is notably smaller** than that observed from the original set to 1hopEZ.
>
> **This suggests that, for certain models, data contamination may pose a significant risk.**
>
> ###  R to W2&Q2&L1
> We conduct a human study on the final generated data, which demonstrates that our framework produces **high-quality dynamic evaluation samples**. We agree that a more fine-grained analysis of the intermediate steps and hop counts would enable more precise quality validation.
>
> We are currently conducting **additional fine-grained human evaluations**; however, due to the limited rebuttal time, we will report these results in the second rebuttal if time permits.
>
> ### R to W3&L1
> Our framework is primarily designed for VQA-format tasks. We are currently conducting additional experiments on VQAv2 to validate its generalization to other standard VQA settings. We will report these results in the second rebuttal if time permits.
>
> **Notably, many multimodal tasks can be reformulated into the VQA format.** Extending our framework to other task formulations is an interesting direction, which we leave for future work.
>
>
> ### R to W4
> Thank you for the suggestion. **Regarding reproducibility**, we conduct ablation experiments using the **open-source Qwen2-VL-72B**, **provide experimental details and prompt examples in the appendix E and F** , and **release the generated dynamic questions as supplementary materials**.
>
> In the revised version, we will further improve the writting of the entire pipeline to facilitate a clearer understanding.
>
> ###  R to Q3
>
> In addition to strict string matching as the evaluation metric, we also adopt **Model-based synonym matching** as an alternative metric **(see Figure 3)**. The results show that, although the absolute accuracy values differ between the two metrics, **the overall conclusions remain consistent**.
>
> Specifically,
>
> **(1) model performance consistently decreases as the hop count increases**
>
> **(2) LLaVA-OneVision exhibits the highest sensitivity to dynamic variations.**
>
> ### R to L1
> Thank you for the valuable suggestions. We will incorporate the discussion of these limitations from the rebuttal into the revised version of the paper.

---

### Official Review · Reviewer_pXoY · 2026-03-12

**Soundness:** 3
**Presentation:** 3
**Significance:** 3
**Originality:** 3
**Overall Recommendation:** 4
**Confidence:** 3

**Summary:**

This paper studies dynamic evaluation for multimodal large language models and argues that static benchmarks suffer from data contamination and performance saturation over time. To address this, the paper proposes KBE-DME, a framework that represents VQA samples as multimodal knowledge graphs and evolves static benchmarks into dynamic ones through triplet re-selection and external knowledge exploration. Based on this process, the method generates new VQA questions with controllable difficulty from existing benchmarks such as OK-VQA and A-OKVQA, and evaluates several representative MLLMs on the evolved test sets. The experiments show that model performance consistently drops as the hop depth increases, and the paper further provides statistics and human evaluation to support the quality of the generated data.

**Compliance With Llm Reviewing Policy:**

Affirmed.

**Final Justification:**

The authors have addressed my concerns, and I have no questions.

**Key Questions For Authors:**

1. The main novelty is moderate. The method is largely a benchmark construction and transformation framework, and the technical contribution on the modeling side is somewhat limited.

2. The whole framework depends heavily on a strong MLLM as the dynamic generation model, including triplet extraction, filtering, exploration, and question generation. This raises concerns about generation bias and reproducibility.

3. The claim that a performance drop may indicate data contamination is not fully convincing. A stronger analysis would be needed to separate contamination effects from simple difficulty increase.

4. The evaluation is still limited to two VQA-style benchmarks, so the claim of broad generalizability to diverse multimodal benchmarks is somewhat overstated.

5. The writing needs improvement. There are many grammar issues, awkward phrasing, and several typos, which hurt clarity.

**Limitations:**

see above

**Strengths And Weaknesses:**

1. The paper studies an important problem. Static multimodal benchmarks indeed become less reliable as models improve, so dynamic evaluation is a meaningful direction.

2. The proposed graph formulation is intuitive, and the overall pipeline of extract–explore–express is reasonably clear.

3. The idea of difficulty control through hop-based knowledge expansion is interesting, and the main results show a clear trend of increasing challenge with more hops.

4. The paper includes human evaluation and data statistics, which help support the quality of the generated benchmark.

---

> ### Author Rebuttal · Authors · 2026-03-31
>
> Thanks for your careful and valuable comments. We will address your concerns point by point.
>
> ### R to Q1
>
> We propose a **holistic dynamic evaluation framework**, rather than a simple benchmark construction or transformation pipeline.
>
> We model VQA problems as **graph structures** and perform dynamic evaluation via controlled graph transformations. This design preserves the **quality of original benchmarks** while enabling **consistent conclusion under different dynamic evaluation settings**, which cannot be achieved by simple data augmentation.
>
> Our framework integrates graph representation, difficulty definition, and dynamic evolution via Re-Selection and Exploration, all of which are essential. A valid dynamic evaluation must satisfy three properties: (1) **controllable difficulty**, (2) **dynamic**, and (3) **conclusion convergence under different dynamic settings**. These can not be easily achieved with naive data transformation.
>
> Specifically, **we define difficulty based on graph structure (e.g., number of edges), enabling controllable difficulty validated by both model and human evaluations (in Figures 3,4,5 and Table 2).** At the same time, evaluations under different dynamic settings yield **consistent conclusions** **(Figures 4 and 5 and extra experiments with Gemini** [due to space limitation, **details can be seen in our response to Reviewer zvvE, R to L1])**, demonstrating robustness and convergence.
>
> ### R to Q2
> **(1) Bias.** We mitigate potential bias by multiple filtering stages in framework. In addition, we employ different models (e.g., GPT-4o and Qwen2-VL-72B and additional Gemini) as dynamic generation models, and observe no significant model-specific bias and inconsistent conclusion when each is used as the generator **(see Figures 4 and 5 and the extra experiment)**.
>
> Moreover, human verification (**Table 2**) shows that the generated triplets achieve **96.8% accuracy**, meeting our quality requirements.
>
> **(2) Reproducibility.** We use the **open-source Qwen-2-VL-72B** as a dynamic generation model, and release the prompts used in our framework in the **appendix F**. Experimental details are further described in **Section 4.1 and appendix E**. We also **provide the dynamically generated 1–3 hop test sets as supplementary materials** to facilitate reproduction and verification.
>
>
> ###  R to Q3
> To further investigate the factors contributing to performance degradation, we conduct a **preliminary analysis in the rebuttal**. In the paper, we define difficulty in a **relative sense**, where the 1-hop expanded dataset is overall more challenging than the original dataset.
>
> Here, for **preliminary analysis**, we additionally use **hop count as a proxy for absolute difficulty** to enable a more fine-grained comparison. **Specifically, we stratify both the original and 1-hop datasets based on the number of edges in the corresponding key subgraphs.**
>
> We select samples from the original dataset with **3-edge key subgraphs** as the control group. From the 1-hop dataset, we select two groups: **1hopEZ** with 3-edge key subgraphs, and **1hopHard** with 4-edge key subgraphs. **To ensure a fair comparison, we randomly sample nearly 1k instances for each group to keep the dataset sizes consistent.**
>
> We then evaluate model accuracy across these three groups, as shown below:
>
>
>
> | Model  | raw(3) | 1hopEZ(3) | 1hopHard(4) |
> | ------ | ------ | --------- | ----------- |
> | GPT    | 46.13  | 33.85     | 32.30       |
> | Gemini | 46.02  | 31.42     | 29.31       |
> | Claude | 49.12  | 33.19     | 30.53       |
> | Qwen   | 42.04  | 26.00     | 23.01       |
> | LLaVA  | 54.09  | 24.45     | 21.90       |
>
> We observe that although the **original set** and **1hopEZ** have the key subgraph with identical edge counts, model accuracy drops substantially on raw to 1hopEZ, **particularly for the open-source model LLaVA.**
>
> In contrast, when comparing **1hopEZ** and **1hopHard**, both of which are dynamically generated, performance further decreases on 1hopHard; **however, the magnitude of this drop is notably smaller than the drop observed from the original set to 1hopEZ.**
>
> This suggests that, for certain models, **data contamination may pose a significant risk**.
>
>
> ###  R to Q4
> Our generalization lies in not imposing constraints on the **specific VQA task formulation**. For instance, NPHardEval is restricted to **algorithmic tasks**, and DyVal focuses on **reasoning tasks**. In contrast, our method is applicable to **VQA-style multimodal tasks** without requiring restrictions on **task scope**. Therefore, we emphasize **task generalization in Table 1**, rather than merely format generalization.
>
> We are currently conducting additional experiments on standard VQA datasets like VQAv2, and will report these generalization results during the second rebuttal period if time permits.
>
> ### R to Q5
> Thank you for your careful reading. We will revise the manuscript to improve clarity and conciseness in the revised version.

---

> > ### Author Rebuttal · Reviewer_pXoY · 2026-04-01
> >
> > Thanks for the rebuttal, most of my concerns are addressed. I would improve my score.

---

> > > ### Author Response · Authors · 2026-04-02
> > >
> > > Thank you very much for your positive feedback. Your constructive comments have helped us improve the clarity and overall quality of the manuscript.

---

### Official Review · Reviewer_zvvE · 2026-03-16

**Soundness:** 3
**Presentation:** 3
**Significance:** 1
**Originality:** 2
**Overall Recommendation:** 3
**Confidence:** 3

**Summary:**

This paper proposes a framework for dynamically evolving multimodal VQA benchmarks to combat data contamination and enable difficulty control. The approach models VQA as a knowledge graph: visual knowledge (G_M), textual knowledge (G_T), and key knowledge triplets (G_K) that link them. Two evolution strategies are proposed: Re-Selection (choosing different key triplets from the same graph) and Exploration (incorporating external knowledge triplets to create novel questions). Difficulty is controlled via hop count in the knowledge graph. Applied to OK-VQA and A-OKVQA. Human evaluation shows 95% of evolved questions rated reasonable, 96.8% triplet correctness, and 97.9% alignment.

**Compliance With Llm Reviewing Policy:**

Affirmed.

**Key Questions For Authors:**

**The "external knowledge" in Exploration is underspecified.**

Where do the new triplets come from? Wikidata? ConceptNet? An LLM? The source and quality of external knowledge directly affects the evolved questions' validity. If sourced from an LLM, generated knowledge can be hallucinated.

**Re-Selection may produce trivially different questions.**

If Re-Selection just picks different key triplets from the same graph, the surface form changes but the reasoning pattern may be very similar. How different are Re-Selected questions in practice? Measuring question diversity (e.g., via embedding similarity) would be useful.

**Limited to knowledge-grounded VQA.**

The approach is applied to OK-VQA and A-OKVQA, which are explicitly knowledge-dependent. It's unclear how this extends to perception-focused VQA (VQAv2, GQA) where the "knowledge graph" formulation is less natural.

**Limitations:**

**Knowledge graph construction quality is the bottleneck.**

The entire framework depends on accurately extracting knowledge triplets from images (G_M) and text (G_T). How are these extracted? If using an LLM/VLM to build the graph, you're using the same models you're trying to evaluate — a circular dependency. The paper must clearly describe and validate the knowledge extraction pipeline.

**Scalability of graph construction.**

Building knowledge graphs for every VQA sample is labor-intensive. How automated is this process? What is the human effort per evolved question? If significant manual curation is needed, the approach doesn't truly "scale" as claimed.

**Hop count ≠ difficulty.**

Using graph hops as a difficulty proxy is intuitive but simplistic. A 1-hop question requiring specialized knowledge may be harder than a 3-hop question connecting common facts. The paper should validate the hop-difficulty correlation empirically (e.g., model accuracy vs. hop count should be monotonically decreasing).

**Strengths And Weaknesses:**

## Strengths to Acknowledge

- Timely problem

Benchmark contamination is a major concern as LLMs/VLMs train on ever-larger web crawls. A principled approach to evolving benchmarks has clear value.

- Elegant graph formulation

Decomposing VQA into knowledge triplets provides a compositional structure that enables systematic manipulation — more principled than ad-hoc perturbation approaches.

- Two complementary strategies

Re-Selection tests robustness to surface variation while Exploration tests genuine generalization to new knowledge — a useful distinction.

- Strong human validation

95%+ on reasonableness, correctness, and alignment is reassuring for benchmark quality.

---

> ### Author Rebuttal · Authors · 2026-03-31
>
> Thanks for your careful and valuable comments. We will address your concerns point by point.
>
> ### R to Q1
> As described in **Line 309 and Sec 4.1**, **we experiment with GPT-4o or Qwen2-VL-72B as the dynamic generation models to expand triplets**, using the prompt illustrated in **Figure 11**.
>
> As described **in the right of Line 257,** during question construction in each hop, we additionally perform  **multiple filtering steps.**
>
> Moreover, **as described in figure 2**, our framework enforces **multi-stage iterative expansion**. If hallucinated knowledge is introduced in intermediate steps, it will be filtered out in next exploration stage. Through our multi-stage multiple filtering process, **we filter out 718, 801, and 779 samples in the three expansions**, respectively..
>
> Finally, **as described in table 2**, we conduct a human study, achieving an **accuracy of 96.8%** on triplets. This result indicates that the **triplets are largely correct**.
>
> ### R to Q2
> **1.** **Re-Selection is performed jointly with the Exploration**.
>
> Re-Selection aims to re-select visual information, which complements the Exploration.
>
> **2.** **Re-Selection itself induces substantial changes to the original question.**
>
> (1) We compute the overlap ratio of key subgraph triplets between before and after Re-Selection, which is **43.85%**.
>
> (2) We measure the **semantic similarity of the questions** between before and after Re-Selection using embedding-based similarity, yielding an average score of **45.22%**.
>
> These results demonstrate that **Re-Selection alone can introduce significant dynamic variations**.
>
> ### R to Q3
> Our **Re-Selection** mechanism is applicable to both perception-focused and knowledge-dependent VQA. **As long as the original question involves visual information, Re-Selection can be used to reconstruct the question.**
>
> The **Exploration** stage incorporates **external knowledge or commonsense reasoning**. Through the combination of them, KBE-DME can generate various questions while preserving the requirement for perceptual understanding.
>
> If time permits, we will include more results on the second rebuttal.
>
> ### R to L1
> As stated in the **right of Line 235**, we employ a **DGM** for triplet extraction, achieving **96.8% accuracy** in all extracted and explored triplets in the human study **(Table 2)**, indicating high reliability.
>
> As described around **right of Line 310**, **we use GPT-4o as the primary DGM and Qwen2-VL-72B for ablation.** Results in **right of Line 376** show consistent conclusions across DGMs, **sugsgesting that model bias and circular dependency do not materially affect our findings.**
>
> To further validate this, we add experiment using **Gemini 2.5 Pro as DGM**. Due to space limits, we report  raw and 1-hop results. Since different DGMs yield different filtered data, we also report raw performance in parentheses.
>
> Due to the unavailability of the Claude 3.5 API, we evaluate Claude 4.5 under Gemini-based DGM.
>
> | DGM  \| Tested | GPT-4o         | Gemini-2.5-pro  | Qwen-2.5-VL     | Claude-3/4.5  | LLaVA-OV      |
> | -------------------- | -------------- | --------------- | --------------- | ------------- | ------------- |
> | GPT-4o               |32.77(47.00)| 30.29(46.38)|24.41(41.24)|32.11(49.36)|23.17(53.19)|
> | Gemini-2.5-pro       |35.35(53.91)|37.11(52.15)|25.78(46.29)|38.87(51.17)|25.78(55.47)|
> | Qwen-2-VL-72B        |29.47(50.00)|26.58(45.26)|27.11(42.89)|30.00(46.84)|20.00(52.11)|
>
> As shown here, although model performance varies across different DGMs, the **relative change with respect to the raw setting remains stable overall**.
>
> We observe that:
>
>  (1) performance consistently decreases on the 1-hop data across all models;
>
>  (2) LLaVA-OneVision still exhibits the largest performance drop.
>
> **This demonstrates that our dynamic evaluation framework consistently mitigates data contamination and saturation issues under different settings, while yielding stable and consistent conclusions, which can not be achieved by traditional static evaluation.**
>
> ### R to L2
> **As stated in the right of Line 235**, we use a **DGM** to extract knowledge graphs from the original VQA questions. **Our framework is fully automated; human evaluation is only conducted to verify the correctness.**
>
> The overall computational cost is reported in Appendix E.
>
> ### R to L3
> While it is possible that some 1-hop questions may be more challenging than certain 3-hop questions, our difficulty control does not rely on the **absolute hop count**. Instead, we adopt a **relative difficulty formulation**: for each instance, the expanded question is expected to be more difficult than its original version.
>
> Therefore, our notion of difficulty is **dataset-level and relative**.  This trend is reflected in the model accuracy reported in **Figures 4 and 5.**
>
> We further validate this **difficulty control through a human study (Appendix D)**, where annotators agree with our difficulty assessment in **84.15%** of the cases.

---

### Decision · Program_Chairs · 2026-04-30

**Decision:**

Accept (regular)

**Comment:**

All reviewers reached a consensus that the proposed multimodal knowledge-graph formulation is elegant or intuitive, noting that decomposing VQA instances into structured triplets provides a more principled and compositional approach to question evolution than superficial perturbations. The hop-based expansion mechanism provides controllable difficulty, and reviewers acknowledged the convincing trend of decreasing model performance as reasoning chains lengthen. Also, the idea of dynamically evolving benchmark (as static benchmark can be potentially saturated and contaminated in reality) is a decent contribution.

Most concerns of two actively involved reviewers are addressed, and they both lean towards weak accept, which is also my recommendation.